# Proximity-Induced Pharmacology for Amyloid-Related Diseases

**DOI:** 10.3390/cells13050449

**Published:** 2024-03-04

**Authors:** Andrea Bertran-Mostazo, Gabrielė Putriūtė, Irene Álvarez-Berbel, Maria Antònia Busquets, Carles Galdeano, Alba Espargaró, Raimon Sabate

**Affiliations:** 1Department of Pharmacy and Pharmaceutical Technology and Physical-Chemistry, School of Pharmacy, University of Barcelona, 08028 Barcelona, Spain; abertran@ub.edu (A.B.-M.); putriutegabriele@gmail.com (G.P.); ialvarezber@ub.edu (I.Á.-B.); mabusquetsvinas@ub.edu (M.A.B.); aespargaro@ub.edu (A.E.); 2Institute of Biomedicine (IBUB), University of Barcelona, 08028 Barcelona, Spain; 3Institute of Nanoscience and Nanotechnology (IN^2^UB), University of Barcelona, 08028 Barcelona, Spain

**Keywords:** amyloid-related diseases, conformational diseases, neurodegenerative diseases, drug discovery, proximity-induced pharmacology, PROTACs

## Abstract

Proximity-induced pharmacology (PIP) for amyloid-related diseases is a cutting-edge approach to treating conditions such as Alzheimer’s disease and other forms of dementia. By bringing small molecules close to amyloid-related proteins, these molecules can induce a plethora of effects that can break down pathogenic proteins and reduce the buildup of plaques. One of the most promising aspects of this drug discovery modality is that it can be used to target specific types of amyloid proteins, such as the beta-amyloid protein that is commonly associated with Alzheimer’s disease. This level of specificity could allow for more targeted and effective treatments. With ongoing research and development, it is hoped that these treatments can be refined and optimized to provide even greater benefits to patients. As our understanding of the underlying mechanisms of these diseases continues to grow, proximity-induced pharmacology treatments may become an increasingly important tool in the fight against dementia and other related conditions.

## 1. Introduction

Amyloid-related diseases are a group of disorders that are characterized by the accumulation and deposition of abnormal protein aggregates, known as amyloids, in various tissues and organs of the body. These diseases can affect different parts of the body, including the brain, heart, kidneys, liver, and other organs. Examples of amyloid-related diseases include Alzheimer’s disease (AD), Parkinson’s disease (PD), Huntington’s disease (HD), type 2 diabetes (T2D), and systemic amyloidosis (SA). The symptoms of these diseases can vary depending on the specific disorder and part of the body that is affected, but they can include cognitive decline, movement disorders, organ dysfunction, and other neurological or systemic symptoms [1,2]. Amyloid-related diseases are a significant area of research and drug development, as effective treatments for these disorders are currently limited. Proximity-induced pharmacology (PIP) and other innovative approaches to drug development are being explored to develop new and more effective treatments for these diseases [3].

PIP is a promising new approach used to develop treatments for amyloid-related diseases, such as AD. The concept behind PIP is to target the underlying mechanism of disease progression by preventing the formation of amyloid plaques in the brain. PIP treatments can take various forms, including small-molecule drugs, peptides, antibody therapies, gene therapies, and chimeric molecules such as PROTACs (proteolysis-targeting chimeras), PhosTACs (phosphorylation-targeting chimeras), AUTACs (autophagy-targeting chimeras), among others. These approaches have shown promise in preclinical studies, and some are now entering clinical trials for various diseases [4].

In this manuscript, we will outline several examples of how PIP treatments offer a promising new approach to treat amyloid-related diseases by targeting the underlying mechanism of disease progression. We will show degradative (i.e., targeted protein degradation, TPD) and nondegradative strategies. In both, ongoing research is exploring the potential of PIP treatments to provide new and effective treatments for patients and families affected by these devastating diseases (see Table 1).

## 2. Proximity-Induced Pharmacology (PIP) Using Degradative Approaches

### 2.1. Molecular Glues

Molecular glues (MGs) are small molecules that insert into a naturally occurring protein–protein interaction, making interactions with both a target protein (protein of interest) and a E3 ligase, leading to the ubiquitination and consequent degradation of the target protein. In the early 1990s, *P. Schrieber* described MGs that could stabilize the interaction between two different proteins (e.g., cyclosporine A and FK506) [5]. However, a crucial milestone in this field was the observation that immunomodulatory drugs (IMiDs, i.e., thalidomide, lenalidomide, pomalidomide) binding to the E3 ligase cereblon promoted the degradation of important non-native substrates, such as ikaros, aiolos, GSTP1, and CK1a. MGs can regulate a variety of biological processes by promoting protein dimerization or colocalization [6]. Although MGs have some characteristics, such as they are monomeric molecules with low molecular weight, have good pharmacokinetic properties, and they can penetrate the blood–brain barrier (BBB), that make them attractive candidates for proximity-induced pharmacology, systematic strategies for the discovery and rational design of new molecules are still lacking [7].

MGs targeting GRB2. In 2021, Qi et al. demonstrated that MG ginsenoside (Rg3) has a neuroprotective effect. This molecule triggers an interaction between the tyrosine kinase receptor (TRKA) and the growth factor receptor-bound protein 2 (GRB2). The ternary complex can activate the extracellular signal-regulated kinase (ERK) cascade in PD models, promoting cell survival while inhibiting the apoptotic program that leads to the massive loss of dopaminergic neurons. Therefore, MGs have great potential as therapeutics for proteinopathies in neurodegenerative diseases [8]. In summary, although the application of MG degraders in the treatment of PD and AD still awaits further research, MGs have significant developmental value for the study of neurodegenerative diseases [6,9].

### 2.2. PROTACs

PROteolyisis TArgeting Chimera molecules (PROTACs) are heterobifunctional molecules composed of two molecules connected via a linker. One molecule is a ligand that binds to a protein of interest (POI), and the other is a ligand of an E3 ubiquitin ligase. These molecules aim to hijack the ubiquitin proteasome system (UPS) by recruiting any POI to an E3 ubiquitin ligase. The formation of the ternary complex promotes the proximity-induced ubiquitination and subsequent degradation of the targeted protein by the proteasome. In contrast to conventional small-molecule inhibitors, PROTACs can be reused, allowing a single molecule to catalyze the ubiquitination of multiple POIs (Figure 1) [10,11].

First-generation PROTACs had peptide-based short motifs as E3 ligase ligands. Peptidic PROTACs have poor cell permeability and low stability in biological systems due to their high molecular weight and labile peptide bonds [12,13]. Even though this first generation of PROTACs had some limitations, they demonstrated the ability to hijack E3 ligases as a new approach for drug development by modulating targeted protein degradation [10].

The later discovery of small-molecule ligands for E3 ubiquitin ligases, including mouse double minute 2 homologue (MDM2), cellular inhibitor of apoptosis protein 1 (cIAP1), cereblon (CRBN), and Von Hippel–Lindau (VHL), allowed for the design of more efficient small-molecule PROTACs. This improvement provided better in vivo stability and distribution of these molecules and raised their potencies from the micromolar to the nanomolar range [14]. To date, a wide variety of PROTACs have been designed and developed for the treatment of concerning conditions such as cancer, immune disorders, neurodegenerative diseases, cardiovascular diseases, and viral infections [15,16]. Given that the impaired clearance of amyloid proteins is a key factor in the pathogenesis of amyloid-related diseases, PROTACs can provide a distinctive approach to these diseases by degrading disease-causing proteins.

Peptide-based tau PROTACs. Tau protein aggregates are a characteristic feature observed in frontotemporal dementia (FTD), AD, and other tauopathies [17]. The accumulation of aberrant forms of tau protein leads to protein aggregation and consequent neuronal death in focal brain areas. In addition, high levels of tau protein can also mediate the toxicity of amyloid-β (Aβ), which is another characteristic hallmark of AD. In healthy neurons, tau promotes the assembly and stabilization of microtubules. However, hyperphosphorylated tau aggregates and dissociates from microtubules, leading to neuronal degeneration. In 2016, Chu et al. developed the PROTAC TH006 to target tau degradation (Figure 2) [18]. This was the first attempt to apply PROTACs to treat neurodegenerative diseases. Chu et al. used the YQQYQDATADEQG sequence as a tau-recognition motif, a short linker consisting of GSGS to increase the flexibility of the PROTAC [19], the ALAPYIP sequence to recognize Von Hippel–Lindau (VHL) E3 ligase, and a poly-arginine (D-Arg)_8_ tail fused to the C-terminus of the PROTAC to facilitate its penetration into cells [13]. Assays using TH006 confirmed that this PROTAC could penetrate inside the cells and induce intracellular tau degradation in a dose- and time-dependent manner in tau-overexpressing Neuro-2a cells, tau-overexpressed human neuroblastoma-derived SH-SY5Y cells, and primary neuron cells derived from the 3xTg-AD transgenic mouse model. When injecting TH006 in vivo, it promoted tau degradation in the brain of 3xTg-AD mice. Moreover, the TH006-mediated degradation of tau reduced Aβ-induced neurotoxicity in the cell cultures mentioned above [5].

Small-molecule-based tau PROTACs. In 2019, Silva et al. synthesized a PROTAC named QC-01-175 (Figure 2) using a known tau positron emission tomography (PET) tracer, 18F-T807, as a tau protein ligand and appended it to the cereblon (CRBN) E3 ligase ligand via a PEG2 linker [20]. This T807-derived compound was able to bind and promote tau clearance in FTD patient-derived heterozygous neurons expressing tau-A152T and tau-P301L in a concentration-dependent manner. Interestingly, when only about 50% of the tau expressed had the A152T or P301L variant, tau clearance in each case was around 70% and 60%, respectively, suggesting that this degrader was also able to target some forms of nonmutant tau in FTD neurons that were also misfolded. On the contrary, the effect of tau degradation on healthy wild-type (wt) neurons was minimal [20]. In 2022, Silva and coworkers used QC-01-175 as a lead compound to discover optimized degraders applying neuronal models from FTD-patient-induced pluripotent stem cells (iPSC) [21]. In A152T and P301L neurons, the second generation of molecules, FMF-06-038 and FMF-06-049, showed a higher degradation of total tau and phosphorylated tau (p-tau S396) up to 60–80% in a short treatment. Also, these optimized degraders showed higher specificity for insoluble protein and promoted a prolonged reduction in tau levels. When changing/replacing the E3 ligase ligand from CRBN to VHL ligands, the results suggested that in human neurons, CRBN has higher activity towards tau ubiquitination for proteasome degradation and might be a better E3 ligase than VHL [5,21].

In 2019, Kargbo published a novel series of tau-targeting PROTACs with a pyridoindole moiety as a tau-binding motif, and small-molecule CRBN and VHL binding moieties were tethered through PEG-based linkers [22]. In a degradation experiment using human tau-P301L and tau-A152T neurons, phosphorylated tau (p-tau) and total tau proteins were successfully eliminated. To verify the potential clinical applications of the compounds, an in vivo pharmacokinetic study was performed in mice, and the capacity of these PROTACs to cross the blood–brain barrier was confirmed [5].

In 2021, Wang et al. reported PROTAC C004019 (Figure 2), which contains a VH032-derived ligand to recruit the VHL E3 ligase and a triazole-based tau ligand connected via a PEG3 linker [23]. C004019 was able to selectively induce tau clearance in HEK293 and SH-SY5Y cells expressing human tau (htau). A single dose and multiple doses of C004019 administrated subcutaneously to 3xTg-AD and htau-transgenic mouse models achieved a sustained tau reduction and alleviated Aβ-induced neurotoxicity in the brain of the animals and improved synaptic and cognitive functions, without showing obvious abnormalities due to tau knockdown [23]. Parallel to these examples, companies like Arvinas Inc. (New Haven, CT, USA), are also developing tau degraders. Arvinas’ tau-targeting PROTAC has demonstrated a reduction in insoluble aggregated tau protein with respect to vehicle treatment in preclinical studies using a murine tauopathy model [24].

Small-molecule-based α-synuclein PROTACs. The misfolding and aggregation of α-synuclein (α-syn) protein into highly ordered amyloid fibrils result in the formation of cytoplasmic Lewy body inclusions, which are pathological characteristics of PD [25]. In 2020, Kargbo reported a patent of six small-molecule PROTACs to target α-syn degradation [26]. These degraders contained 2-(4-*N*-methylphenyl) benzothiazole, 1-benzyl-3-(3-(4-nitrophenyl)allylidene)indolin-2-one, and 3-nitro-10H-phenothiazine as α-syn-binding motifs. Small-molecule ligands binding to CRBN and VHL E3 ligases were coupled using PEG or cyclic amine-based linkers. Degradation activities in HEK293 TREX α-syn A53T cells showed α-syn degradation levels >65% for several designs [26]. Recently, novel small-molecule degraders for α-syn have been developed based on an already known inhibitor of α-syn aggregation: sery384. The specific binding of these molecules to α-syn aggregates has been assessed by computational docking studies, and the degradation efficiency of the designed PROTAC molecules has been evaluated in vitro. The most potent compound, with a DC_50_ of 5 μM, reduces α-syn aggregates in a time- and dose-dependent manner, lowering the toxicity induced by α-syn aggregates [27].

Small-molecule-based TDP-43 PROTACs. The cause of amyotrophic lateral sclerosis (ALS) is not yet fully understood, but TAR DNA-binding protein (TDP-43) aggregates are considered one of the key pathogenic agents since they are found in 97% of all ALS patients [28]. In 2023, Tseng et al. assessed their newly developed PROTAC JMF4560 (Figure 2) to target C-terminal TDP-43 (C-TDP-43) aggregates in Neuro-2a cells and *C. elegans* [29]. This molecule was constructed by pomalidomide as a CRBN binder, a PEG4 linker, and a benzothiazole aniline (BTA) as a C-TDP-43 binder. They demonstrated that JMF4560 significantly reduced C-TDP-43 aggregates and alleviated C-TDP-43-induced toxicity in Neuro-2a cells without affecting endogenous full-length TDP-43. Furthermore, it was revealed that JMF4560 could reduce the compactness and population of C-TDP-43 toxic oligomers. In the *C. elegans* model, JMF4560 could reduce C-TDP-43 aggregates in the nervous system, and it exhibited beneficial effects in terms of motility [29].

Small-molecule-based GSK-3β PROTACs. Glycogen synthase kinase 3 beta (GSK-3β) is a multifunctional class of serine/threonine protein kinases. Studies have shown that GSK-3β can promote AD by increasing the levels of phosphorylated tau protein (p-tau) and Aβ. This can induce a proinflammatory effect that leads to neuronal loss [30,31,32]. In 2021, Jiang et al. reported a GSK-3β-targeting PROTAC called PG-21 (Figure 2). This compound recruits GSK-3β protein to a ligand that binds to the CRBN E3 ubiquitin ligase. Assays have shown that this compound can effectively degrade GSK-3β in a dose-dependent manner, inducing over 44% degradation. Moreover, it exhibits a neuroprotective effect by protecting against glutamate-induced cell death in HT-22 cells [16,33]. Recently, a new set of GSK-3β PROTACs was synthetized by linking two different GSK-3β small-molecule inhibitors, SB-216763 and tideglusib, with pomalidomide (as the CRBN binder for E3 ligase recruitment). One of the PROTAC molecules based on the SB-216763 inhibitor demonstrated the capacity to degrade GSK-3β at low micromolar concentrations. Furthermore, it has shown reduced neurotoxicity induced by the Aβ peptide in SH-SY5Y cells and has not exhibited toxicity in neuronal cells below 20 μM treatments [34].

Small-molecule-based LKRR2 PROTACs. Leucine-rich repeat kinase 2 (LRRK2) is a very interesting target for PD and other neurodegenerative disorders. LRRK2 is a multidomain GTPase/kinase that has scaffolding functions. LRRK2 regulates a variety of downstream processes related to lysosomal and mitochondrial function, neuroinflammation, and α-synuclein accumulation. There are data suggesting that a reduction in LRRK2 in the brain could be promising for the treatment of PD [24]. Recently, a PROTAC molecule targeting LRRK2 was developed as an alternative to traditional LRRK2 inhibitors. XL01126 (Figure 2) is described as a rapid and potent degrader of LRRK2 with a DC50 in the nanomolar range and can achieve more than 80% LRRK2 degradation. It is an orally bioavailable molecule and can penetrate the blood–brain barrier. These properties and all the data shown in the study make LRRK2 degrader appear as an attractive probe to target LRRK2 and develop new drugs for PD [35]. Furthermore, Arvinas Inc. company has developed bioavailable LRRK2 PROTAC degraders that can penetrate the brain and distribute around affected regions. These degrader molecules have been optimized to improve selectivity and potency after a single oral administration. LRRK2 PROTAC from Arvinas achieves a 50% durable degradation of LRRK2 protein for 2–3 days before the molecule is cleared from the brain and LRRK2 protein is resynthesized [24].

Small-molecule-based mHtt PROTACs. HD is mainly provoked by aggregates of mutant huntingtin (mHtt) protein in neurons. These aggregates predominantly form due to the presence of an expanded polyglutamine domain in the pathogenic protein. Arvinas company has identified new small-molecule ligands and has used them to develop PROTAC molecules able to induce the potent and selective degradation of mHtt without any effect on wtHtt protein [24]. In addition, Origami Therapeutics company (San Diego, California, USA) is also working on developing mHtt degraders (ORI-113) to remove protein aggregates from the body [36].

### 2.3. SNIPERs: IAP-Based PROTACs

SNIPERs are bivalent degrader molecules that are composed by a specific E3 ubiquitin ligase, an inhibitor of apoptosis (IAP) protein, a linker, and a ligand for a specific target protein, leading to target protein ubiquitination and degradation. Specific and nongenetic inhibitors of apoptosis proteins (SNIPERs) are widely used to induce the degradation of multiple protein targets [37,38,39]. The first generation of SNIPERs failed due to autoubiquitination and self-degradation. However, novel series of SNIPERs circumvent these undesired processes. SNIPERs have worked well not only in cancer targets (e.g., targeting estrogen receptor α (ERα), bromodomain-containing protein 4 (BRD4)) but also in immune diseases and neurodegenerative disease targets, such as mutant tau and Htt [37,39,40]. Currently, SNIPERs have demonstrated the ability to degrade more than 20 proteins, including undruggable targets [37]. SNIPERs offer several advantages compared to small-molecule drugs. Nevertheless, more studies are needed to assess SNIPER absorption, distribution, metabolism, excretion, toxicity, and other properties [40].

mHtt SNIPER. Several SNIPERs have been developed in the last few years for amyloid-related diseases such as mHtt. Various approaches have been developed and described to target toxic mHtt aggregates, but they have some drawbacks, such as low permeability and poor metabolic stability and selectivity [41]. Despite numerous efforts, there is currently no effective clinical treatment available for HD due to unclear understanding of the mechanisms of action of aggregation. In 2017, Minoru Ishikawa et al. successfully targeted the aggregation of mHtt using small-molecule SNIPERs. They developed SNIPER-48 and SNIPER-49 degraders by combining benzothiazole aniline (BTA) and phenyldiazenyl benzothiazole derivative (PDB), small-molecule probes binding to mHtt [42], with bestatin through appropriate linker molecules. This dose-dependent degradation indicates that SNIPER-48 and SNIPER-49 (Figure 2) are effective regulators of mHtt protein levels in HD patients’ fibroblasts, making them encouraging candidates for the treatment of HD patients. Unfortunately, the authors did not extensively address the drug-like properties of SNIPER molecules, such as membrane permeability and metabolic stability, reflecting the need for improvement in some of the pharmacokinetic properties to exploit the therapeutic potential of these molecules [40]. Yamashita et al. also found these SNIPERs to be effective in disrupting mutant ataxin-3 and ataxin-7 in patients with spinocerebellar ataxia and atrophin-1 in patients with dentatorubral pallidoluysian atrophy. These results suggest that SNIPERs can efficiently disrupt the intersecting β-sheet structure prevalent in some neurodegenerative diseases and should be useful both as research tools and as candidates for the treatment of several neurodegenerative diseases [6,43].

### 2.4. AUTACs and ATTECs

Targeted protein degradation strategies, exemplified by PROTACs, have limitations because many intracellular materials are not substrates for proteasomal clearance. Autophagy targeting chimera molecules (AUTACs) are a novel targeted-clearance therapeutic approach that utilizes the autophagy pathway, the major intracellular degradation mechanism, to selectively degrade disease-causing proteins. Autophagy plays a critical role in maintaining cellular homeostasis, including clearing damaged or misfolded proteins. Therefore, the use of AUTACs to restore proper autophagy function may provide therapeutic benefits. AUTACs induce the polyubiquitination of the target protein and the addition of a degradation tag (guanine derivatives) that makes the target protein sequester into autophagosomes. The autophagosomes then fuse with lysosomes, where the target protein is degraded by the lysosomal hydrolases. AUTACs are mainly composed of three parts: the ligand of the target protein, the linker, and the degradation tag guanine derivative [6].

Neurodegenerative diseases, such as AD, PD, FTD, HD, and ALS, have mutations that affect the autophagy mechanism. Characteristic protein aggregates have been observed in autophagy-deficient neurons in animal models [44]. In neurodegenerative diseases, AUTACs have been designed to selectively target and degrade abnormal proteins, such as tau and α-syn, implicated in the development and progression of these diseases. AUTACs capture these proteins resistant to entering the autophagy system, directing them into it [45]. In the same line, there is another degradative approach that also uses autophagy to induce protein degradation. Autophagosome-tethering compounds (ATTECs) are small molecules that bind simultaneously to the target protein and the autophagosome protein LC3, inducing target protein degradation via the autophagy pathway [8].

Tau AUTACs. Ji and co-workers developed an AUTAC that can mediate the targeted degradation of oligomeric species of aggregation-prone proteins and confirmed the therapeutic efficacy of a misfolded protein-targeting AUTAC in a brain-specific murine model expressing human mutant pathological tau [46].

α-syn AUTACs. The accumulation of misfolded α-syn protein leads to the dysfunction and degeneration of PD neurons. Although the degradation of α-syn using autophagy machinery has not been experimentally proven yet, the development of AUTAC molecules targeting α-syn accumulation has a huge potential for PD treatment. In 2010, a peptide-based molecule was designed to target α-syn aggregates, triggering them into lysosomal degradation. The molecule consisted of a β-syn 36 peptide able to bind α-syn and a peptide able to bind a chaperone protein that triggers α-syn for degradation [8,47]. Recently, a new AUTAC molecule has been developed to target α-syn aggregates for lysosomal degradation. ATC161 is a chimeric compound that uses Anle318b as a ligand to bind α-syn aggregates. The other domain of the molecule recruits and activates the autophagy receptor SQSTSM1/p62 (sequestosome 1). This compound can direct α-syn aggregates into phagophores and consequent lysosomal degradation. Degradation is achieved at 100–500 nM of ATC161 without the off-target degradation of monomeric α-syn. This effect has been demonstrated in vivo using PD mice models. The oral administration of ATC161 reduces α-syn aggregates and their propagation across the brain, alleviating in this way the effects provoked by α-syn accumulation, such as glial inflammatory responses and the decrease in locomotive activity [48].

mHtt ATTECs. The huge protein aggregates related to mHtt are resistant to proteasome degradation; thus, the autophagy–lysosomal pathway is a promising alternative to eliminate them [7]. In 2019, Lin et al. identified the first examples of mHtt-LC3 linker compounds tethering mHtt degradation via the lysosomal pathway. The four small molecules found (10O5, 8F20, AN1, and AN2) demonstrated the ability to interact and reduce selectively mHtt without any effect on wtHtt in cultured primary cortical neurons [9]. Furthermore, the efficacy of the compounds was probed in in vivo experiments using HD-knock-in mouse models. The molecules showed good penetration of the BBB and a remarkable reduction in mHtt in the cortices of HD mice [5,6,8].

### 2.5. Chaperone-Mediated Autophagy (CMA) Molecules

The chaperone-mediated autophagy process differs from other autophagy-related proximity-induced approaches because it does not require vacuole formation to trigger protein degradation. The CMA approach mediates protein degradation via the heat shock homology 70 kDa chaperone (Hsc70) that recognizes the KFERQ-like motifs of cytosolic proteins. This complex then interacts with the lysosomal-associated membrane protein (LAMP2A), enhancing its oligomerization and leading to the transfer of the target protein to the lysosomal cavity and consequent protein degradation [6].

CMA targeting Htt. In 2010, the degradation of mHtt by the chaperone-mediated autophagy pathway was successfully achieved for the first time using a peptide-based molecule. The authors designed peptide-based molecules comprising two polyglutamine-binding peptides able to interact with mHtt and two different Hsc70 moieties. These molecules demonstrated their potential for therapeutical approaches in neurodegenerative diseases since they triggered the polyQ protein into degradation via chaperone-mediated autophagy without affecting the levels of wtHtt [49].

CMA targeting α-syn. In 2014, a molecule targeting α-syn degradation via chaperone-mediated autophagy (CMA) was developed. This molecule was peptide-based and consisted of three different motifs: a cell membrane-penetrating peptide (TAT-CPP, sequence: YGRKKRRQRRR), an α-syn motif (α-syn-BM, sequence: GVLYVGKSTR), and a chaperone targeting motif (CMA-TM, sequence: KFERQKILDQRFFE). The interaction between the peptide-based molecule and the target protein α-syn leads the target protein to the lysosomal proteolytic machinery for degradation. Different peptide-based molecules were synthesized, and they demonstrated a specific reduction in α-syn levels in cultured neurons [5].

CMA targeting Aβ aggregates. Bifunctional molecules can also be used to induce chaperone-mediated protein degradation. For example, bifunctional molecules binding simultaneously to the FK506 chaperone-binding protein and to Aβ aggregates. These molecules are designed using amyloid ligands such as Congo Red (CR) attached to synthetic ligands for FKPB. Several experiments indicate that Aβ fibril formation can be inhibited by recruiting chaperones to induce aggregate formation, showing a promising therapeutic potential for AD [50].

### 2.6. Hydrophobic Tags (HyTs)

HyTs involve the use of hydrophobic tags to specific target proteins, leading to their degradation via the proteasome. Crews and colleagues first described the “hydrophobic tagging (HyT)” strategy, a mechanism whereby a protein attached with a hydrophobic tag can mimic a misfolded protein. A typical HyT molecule consists of two components: one is an identifying peptide or small-molecule chemical ligand that specifically binds to the POI, and the other is a hydrophobic tag that mimics the state of the misfolded protein, thereby inducing the degradation of the POI by the recruitment of chaperones or directly by the proteasome [51]. In principle, HyTs are suitable for any protein as they only need to add a low-molecular-weight hydrophobic group to the target protein [52]. Hydrophobic tags have also been reported to have low toxicity [6].

Tau HyT. In 2017, a HyT molecule targeting tau protein degradation was developed. It was composed of three functional molecules: a hydrophobic tag, a tau recognition group, and a cell-permeable peptide group. Both in vitro and in vivo experiments were performed to confirm the specific degradability of the hydrophobic tag with respect to the tau protein. The degradation of the target tau protein by HyT was demonstrated to be dependent on the proteasomal pathway. Unlike PROTAC technology, HyTs do not distinguish normal tau from the abnormal one or polymerized tau from the dissolved one [52].

TDP-43 HyT. It has been shown that the concentration of the monomeric pathogenic protein TDP-43 has a significant impact on the aggregation process of ALS, among other neurodegenerative disorders. The higher the concentration of TDP-43 monomer, the faster the aggregation [53]. Furthermore, imbalances in TDP-43 concentration may be a common feature of TDP-43-induced pathogenesis, and lowering the TDP-43 concentration may be a potential therapeutic intervention. The hydrophobic tagging approach was effective at robustly and specifically degrading TPD-43 [54]. Gao et al. designed D4, a multifunctional peptide-like HyT. The di-adamantyl chemical building block was chosen as the hydrophobic motif, the KGSGS sequence as the linker, and the EDLIKGISV sequence as the TDP-43 binding motif, also the GRKKRRQRRR sequence was also used as the cell-penetrating peptide (CPP) motif. This HyT molecule exhibited the highest ability to promote c-terminal TDP-43 (C-TDP-43) degradation both in vitro and in vivo. D4 has also been shown to be able to enter cells in a short time and to have relatively low cytotoxicity. These results show that the use of HyT to induce the degradation of TDP-43 is an effective potential strategy for the treatment of, for example, ALS [54].

### 2.7. Targeted Protein Removal by an In Situ Method (TRIM-Away)

Another recently developed rapid technology for protein degradation is TRIM-Away. The key component in this process is the intracellular E3 Ubiquitin ligase TRIM21, which binds to the Fc-region of the antibody and is ubiquitinated to the point that it is degraded by the proteasome, along with the bound antibody and its specific target [55]. Administration of the antibody into single cells can be performed by microinjection or into cell cultures by electroporation without altering the level of genome or mRNA expression [49,56,57]. The major advantage of the TRIM-Away approach is that it is rapid and permits the observation of phenotypic changes in the cells, while limiting the occurrence of compensatory mechanisms. However, this approach relies on the utilization of antibodies that exhibit high specificity and do not interact with other intracellular proteins [57].

TRIM family proteins (TRIMs) are characterized by the fact that they are mainly composed of three domains: the RING finger domain (R), the B-box domain (B), and the coiled-coil domain (CC), though not all TRIM proteins have a RING domain. In 2003, it was found that many TRIM proteins are involved in the ubiquitination process and exhibit E3 ligase activity. Many human TRIM proteins are involved in various ubiquitination processes. Several studies have reported that many TRIM family proteins play a significant role not only in the progression of various cancers but also in noncancerous diseases such as rare genetic disorders, neurodevelopmental disorders, and cardiac disorders, making TRIMs a significant choice for further studies in the third decade of E3 ubiquitin drug discovery [58].

Tau TRIM-Away. TRIM21 has tau hyperphosphorylation neutralizing activity, and, together with known intracellular mediating mechanisms such as the molecular unfolding enzyme, the valosin-containing protein (VCP), and the proteasome, TRIM21 takes the tau protein into proteasome-dependent degradation pathways. Since the tau monomer is rapidly degraded before tau aggregates are formed, TRIM-Away technology avoids the inability of the proteasomal pathway to degrade the aggregated protein [6]. Although it is difficult to affect protein aggregation, which is a common drawback of protein-targeted degradation technology that relies on the ubiquitin proteasome system (UPS) pathway, the effect of Trim-Away on the TRIM21 protein degradation of tau protein occurs before the aggregation of tau protein. It is hoped that Trim-Away, as a treatment for Alzheimer’s disease, may have a better preventive effect [6].

Htt TRIM-Away. The pathogenic Htt protein has also been cleaved in vivo using TRIM-Away technology [55]. It was found that the longer the Htt polyQ, the more TRIM21 molecules are involved in the Htt degradation process. The length of the polyQ is related to HD pathogenesis and severity, and the level of degradation correlates with the number of antibody bindings. In juvenile HD (jHD), where patients develop symptoms before the age of 21 years, there is a negative correlation between the length of polyQ repeats and the age of onset of disease. TRIM-Away’s specific recognition of polyQ repeats, in particular, the efficient recognition and subsequent degradation of polyQ longer than 39 glutamine residues, suggesting that TRIM-Away may help in the treatment of jHD. In other words, TRIM-Away technology allows for the targeted degradation of pathogenic Htt proteins by specific recognition of the length of the polymeric protein [55].

## 3. Proximity-Induced Pharmacology (PIP) Using Nondegradative Approaches

### 3.1. PhosTACs and DEPhosphorylation TArgeting Chimera (DEPTACs)

Protein phosphorylation plays a crucial role in regulating numerous aspects of cell biology, and dysregulated protein phosphorylation is associated with many diseases. PhosTACs and DEPTACs represent a novel class of heterobifunctional molecules that works by recruiting phosphatases to target proteins, thereby facilitating targeted protein dephosphorylation. The link between protein phosphorylation and the development of some neurodegenerative diseases is widely known [59].

Tau DEPTAC. Zheng et al. developed a DEPTAC that specifically facilitates the binding of the Bα-subunit of phosphatase 2 A (PP2A-Bα), the most active tau phosphatase in the brain, to p-tau. Zheng’s tau DEPTAC not only promoted tau dephosphorylation at multiple sites with improvements in neuronal morphology and at function levels but also facilitated tau elimination [60].

Tau PhosTAC. Crews’ laboratory developed a tau PhosTAC that can achieve a similar effect as Zheng’s tau DEPTAC, indicating that tau PhosTAC is also a promising approach for treating tauopathies, including AD [61]. In their study, Crews and co-workers applied PhosTAC technology to form ternary complexes that recruit the primary native tau phosphatase, PP2A, to tau, resulting in robust dephosphorylation of the protein and enhancing tau elimination. This tool represents a significant advancement in selectively manipulating protein phosphorylation and dephosphorylation and has the potential to be utilized as both a research and therapeutic tool [61].

### 3.2. Protein–Protein Interaction (PPI) Stabilizers

Small molecules can also induce protein–protein interactions, such as MGs, but without exhibiting a degradative outcome of the POI, since any protein involved in the binding complex is an E3 ligase [62]. The stabilization of concrete targets could also have therapeutical benefits in neurodegenerative disorders.

PPI stabilizers of 14-3-3 protein. 14-3-3 is a highly expressed protein in the central nervous system (CNS). Deletion of the gene encoding 14-3-3 causes Miller–Dieker syndrome (MDS), a congenital disorder with abnormal brain development. Furthermore, disruption of 14-3-3 expression has also been related to PD pathogenesis. Therefore, drugs targeting 14-3-3 could have a great potential for treating neurodegenerative disorders. Some studies suggest that 14-3-3 overexpression may be useful in the recovery of dopaminergic neurons, whereas its inhibition induces neuron loss in a PD mouse model [63]. Kaplan et al. identified 3 hit molecules from 30 analogs derived from the 14-3-3 PPI modulator fusicoccin-A (FC-A). These compounds could potentially stimulate neurite outgrowth and axon regeneration in vitro. These modulators seem to stabilize some of the 14-3-3 interactors and induce the destabilization of others [9].

PPI stabilizers of LC proteins. Conformational changes in Ig light chains (LCs) can induce LCs’ aggregation and provoke a progressive degenerative disease known as light-chain amyloidosis (AL). Chemical stabilizers of LC proteins have been developed and they have demonstrated the capacity to stop the conformational changes in LCs and the consequent aggregation cascade. These chemical stabilizers bind to conserved residues of LC proteins and stabilize their nontoxic structure [64].

## 4. Perspectives and Future Directions

Today, neurodegenerative diseases, specifically those concerning amyloids, lack effective cures, and the currently available treatments, such as small-molecule drug therapy or immunotherapy, only alleviate symptoms. Various therapeutic approaches are being developed to treat neurodegenerative disorders, but they have some limitations, such as low brain penetration and/or off-target effects. In this scenario, proximity-induced drug discovery approaches offer a new way of removing protein aggregates that accumulate in the brain and cause neurodegeneration, leading to several diseases. However, as shown in this revision, there are some limitations to overcome to bring these molecules into the clinic. In this sense, work is ongoing to continue to develop these techniques for clinical use.

PROTACs and SNIPERs hold promising potential in the combination of neurodegenerative diseases. Several modifications are being studied to explore a wide range of features that can improve PROTACs and allow them to be used in concrete conditions. For instance, noncleavable linkers connecting the E3 ligase ligand and the POI ligand could allow irreversible protein degradation, cleavable linkers (by enzymes or chemicals), or photo-switchable PROTACs (that use a photo-switchable ligand that can be activated by light) to permit the spatiotemporal control of protein degradation or trivalent PROTACs that could bind into two different binding sites or two POIs simultaneously, resulting in increased potency and selectivity. All these examples that are under study are viable systems that could be potentially used for amyloid-related diseases. Nonetheless, there are still some limitations that PROTAC technology must overcome to be used as a therapeutic for amyloid-related diseases. The molecules must penetrate the BBB, and not all of them may be able to do it. Moreover, PROTACs must target proteins localized on specific brain regions without affecting proteins in other tissues to avoid undesired side effects. The potency and durability of the PROTAC effect must be evaluated carefully in each case, but the simultaneous usage of PROTACs with other technologies could be a promising option [65]. One of the main limitations in developing PROTACs is the lack of E3 ligase ligands. Only a few E3 ligases have been used for targeted protein degradation approaches. However, targeting E3 ligases with a high expression or exclusive expression in the brain could add another layer of selectivity to these PROTACs. This is a strategy that should be developed in the future. In parallel, PROTAC molecules are only able to degrade intracellular targets, that limits somehow the application of these approaches in amyloid-related diseases.

Since PROTACs and SNIPERs are limited to use in the ubiquitin proteasome system to achieve protein degradation, ATTECs and AUTACs emerge as alternative approaches that direct protein degradation by inducing proximity between the POI and components of other degradative pathways, such as autophagy, that do not rely on the ubiquitination of the target protein. Alternatively, HyT does not rely on E3 ligases to achieve protein degradation since it recruits the POI directly to the proteasome. Again, ATTECs, AUTACs, and HyTs exclusively degrade intracellular targets.

LYTACs represent a highly promising approach for the targeted degradation of specific proteins through the lysosomal degradation pathway. In contrast to the PROTAC approach, LYTACs exhibit the capability of targeting extracellular and membrane-associated proteins by utilizing conjugates that bind both a cell-surface lysosome-shuttling receptor and the extracellular domain of the protein of interest [66]. An LYTAC molecule is formed by fusing two essential components: a binder for the target protein and a ligand for a cell surface lysosomal-targeting receptor (LTR). An endogenous LTR possesses the ability to bind extracellular glycoproteins and facilitate their transportation to lysosomes. The target-binding moiety can encompass antibodies or a small synthetic peptide [66]. LYTAC technology has enormous potential to be used as a pharmacological approach in neurodegenerative diseases since protein aggregates are usually found in the extracellular space of brain tissue. Moreover, the lysosome pathway allows for the degradation of larger protein aggregates, whereas the proteasome does not. LYTACs targeting mHtt are being explored. In this scenario, the LYTAC molecule can bind simultaneously to mHtt and the LTR, leading to the internalization of mHtt protein into the cells via endosomes and later degradation by lysosomes. Degradation of the mHtt protein would reduce the ability of mHtt protein to infect new cells and slow down the spread of mHtt aggregates in the brain [49]. The LYTAC approach can be extrapolated to also work in other pathogenic protein aggregates that accumulate extracellularly, such as the Aβ protein.

TRIM-Away is also an emerging protein degradation technology with great potential for development in neurodegenerative diseases research and treatment. It can be applicable to a wide range of cell types and induces immediate and rapid degradation of the POI. One disadvantage of TRIM-Away as a potential treatment approach is the difficulty in delivering extracellular antibodies to nerve cells, as methods such as injection, transfection, and electroporation are not feasible in the living brain [49]. In addition, the large size of the TRIM-Away molecule results in low penetration into tissues such as the blood–brain barrier or cell membranes, making some epitopes inaccessible due to steric hindrance [49,56]. Furthermore, the exogenous administration of TRIM21 may have side effects on treated cells [56]. So, a new functional delivery method is needed to allow TRIM-Away to enter the brain and nerve cells [49]. While TRIM-Away has shown great potential, it has not yet been successfully applied to the development of novel therapeutic approaches related to protein degradation [49].

Finally, the use of proximity-induced nucleic acid degraders could be a promising option for treating pathologies characterized by protein misfolding, such as AD, HD, and T2D. This is because, in the human genome, most of the RNA transcripts are not translated into proteins. Therefore, there could be many RNA species that are relevant to the disease, making the degradation of RNA an attractive approach [67]. The first attempts to degrade RNA using small-molecule RNA binders were made through RIBOTACs, which are composed of a combination of a small-molecule RNA binder and a ribonuclease ligand. However, RIBOTACs have limitations, such as their dependency on the endogenous concentrations of ribonuclease L., which is not evenly expressed across different tissues. Additionally, RIBOTACs can negatively affect the physiochemical properties of ligands due to their large molecular weight [67]. To address some of the issues associated with RIBOTACs, a new strategy has been proposed for targeting RNA-related diseases using small molecules called proximity-induced nucleic acid degraders (PINADs). Recently, researchers have proposed using PINADs to combat SARS-CoV-2 and potentially other RNA-related diseases. This innovative approach offers the potential to selectively destroy specific RNA species associated with disease pathology. PINADs comprise three different components: a small molecule that binds to RNA, a long flexible linker that allows the molecule to reach multiple positions on the targeted RNA, and an imidazole group that mimics the RNA-degrading moiety present on many ribonucleases [67]. Amyloid-related diseases can also be targeted by RIBOTACs and PINADs.

In summary, extensive efforts are underway to devise novel pharmacological strategies for addressing the unmet clinical needs associated with amyloid-related conditions. With no existing treatments available, and the number of affected patients steadily rising, PIP emerges as a potential solution. It presents a diverse array of approaches that could potentially overcome some of the limitations associated with the current strategies.

## Figures and Tables

**Figure 1 cells-13-00449-f001:**
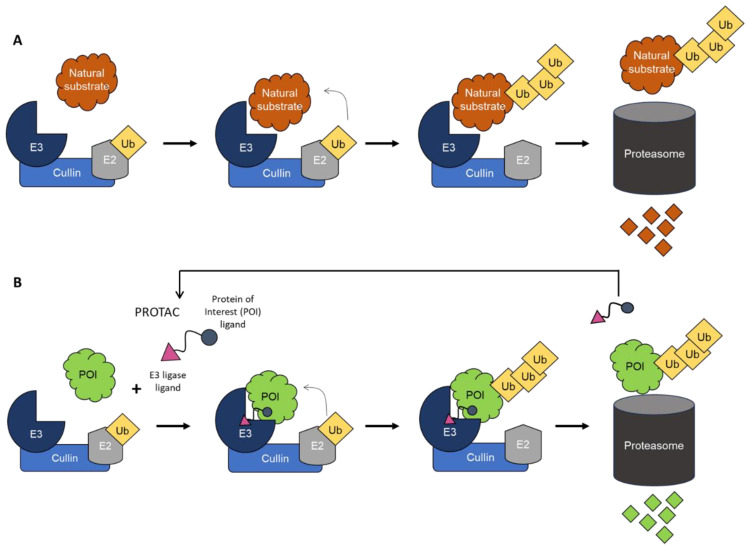
(**A**) Natural pathway of ubiquitin–proteasome system controlling protein degradation and turnover. (**B**). PROTAC-mediated degradation of the protein of interest (POI) by the ubiquitin–proteasome system.

**Figure 2 cells-13-00449-f002:**
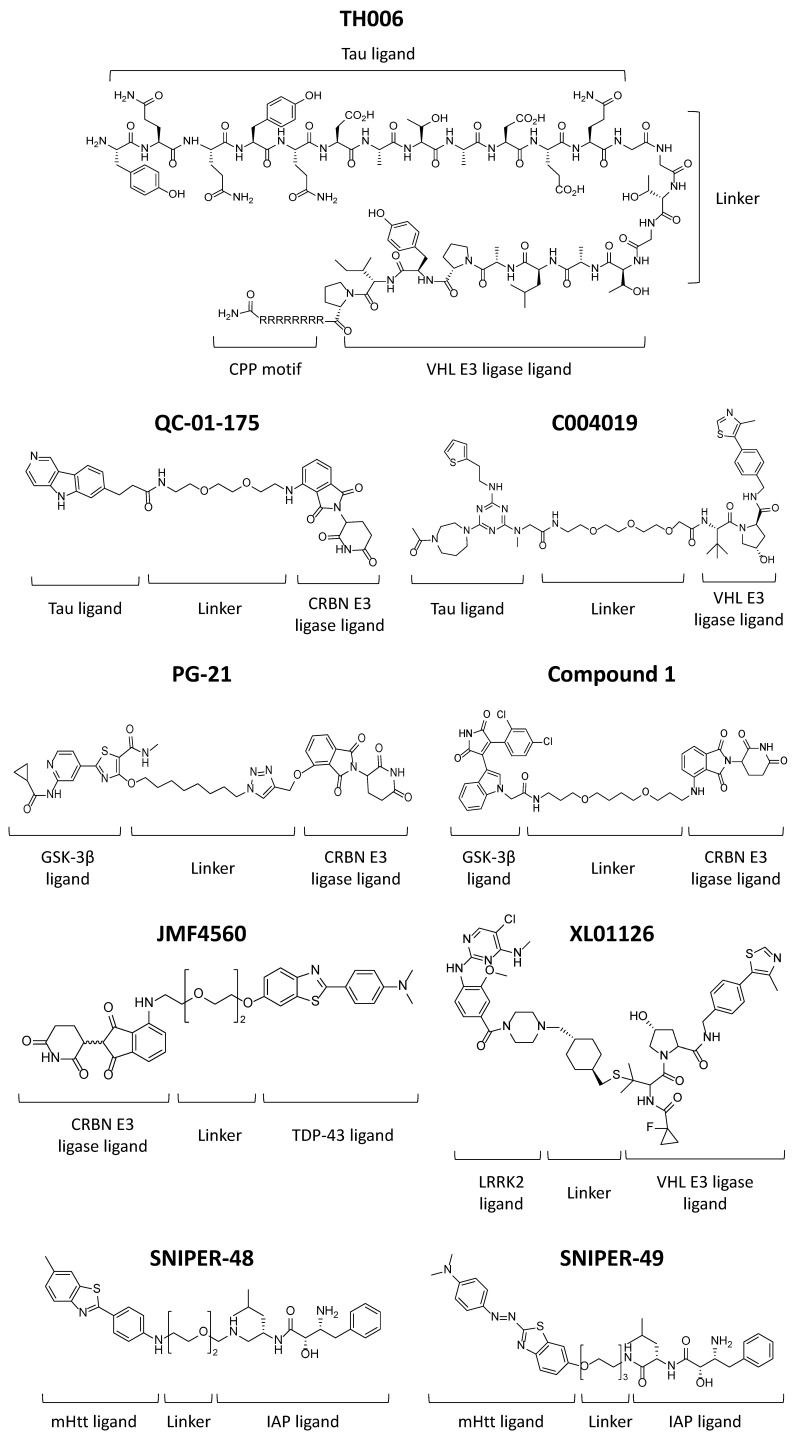
Chemical structures of described PROTACs and SNIPERs.

**Table 1 cells-13-00449-t001:** PIP degradative approaches.

PIP Degradative Approach	Concept	Limitations	Target Proteins Tested Relevant for NDs
Molecular glues (MGs)	Small molecules able to stabilize the interaction between an E3 ligase and a POI, leading to POI polyubiquitination and degradation.	Lack of rational design for novel molecules.	GRB2
PROTACs	Heterobifunctional molecules that put in close proximity a target POI and an E3 ligase, triggering POI ubiquitination and degradation by the proteasome.	Molecules with poor drug-like profile. Lack of well-developed ligands to exploit different E3 ligases.	Tau, α-synuclein, TDP-43, GSK-3β, LKRR2, Htt
SNIPERs	Bivalent degraders (same as PROTACs) that use a specific E3 ubiquitin ligase: IAP E3 ligase.	Unique E3 ligase used. Additionally, molecule absorption, distribution, metabolism, excretion, and toxicity have not been properly assessed.	Htt
AUTACs and ATTECs	Bivalent (AUTACs) or monovalent (ATTECs) degraders that trigger protein degradation through the autophagy/lysosomal pathway.	Lack of understanding about the mode of action and potential off-target effects provoked by hijacking the lysosomal pathway.	Tau, α-synuclein, Htt
Chaperone-mediated autophagy (CMA) molecules	Protein degradation mediated via the Hsc70 chaperone that can interact with LAMP2A and transfer the target POI into the lysosomal cavity to induce POI degradation.	Low molecule stability and poor molecule delivery efficiency.	Htt, α-synuclein, Aβ aggregates
Hydrophobic tags (HyTs)	Hydrophobic tags attached to POIs can mimic the state of misfolded proteins, leading to POI degradation.	Poor biological activity that seems to depend on the length and composition of the hydrophobic tag. New tags and better understanding of the mechanism of action are needed.	Tau, TDP-43

## Data Availability

No new data were created or analyzed in this study. Data sharing is not applicable to this article.

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
