# Peer review of "Proximity-Induced Pharmacology for Amyloid-Related Diseases"

_cells, 2024, doi:10.3390/cells13050449_

Round 1

Reviewer 1 Report

Comments and Suggestions for Authors

Amyloidosis is an umbrella term that describes a group of diseases caused by abnormal deposition of beta-sheet rich fibrillar protein aggregates in various tissuesIn this manuscript, the authors reviewed and discussed the development of proximity-induced pharmacology as a therapeutic approach for amyloid-related diseases, highlighting the recent advances of this strategy in developing new treatments for amyloid-related diseases. This is a well written review, I just have two comments to be addressed.  

1.    In PIP using degrative approaches, it would be helpful to add a table summarizing the concept and limitation of each approach, and the target proteins that have been tested so far, any other useful details for the reader. 

2.    In page 2, line 83, there is a typo error, “n” should be “In”

Author Response

Thank you for your thorough review. We have incorporated all your comments and suggestions.Here are our responses, point by point.

  1. In PIP using degrative approaches, it would be helpful to add a table summarizing the concept and limitation of each approach, and the target proteins that have been tested so far, any other useful details for the reader.

Thanks for this suggestion. We have added a new table summarizing all PIP using degrative approaches.

  1. In page 2, line 83, there is a typo error, “n” should be “In”

Thanks again. We have corrected this typographic mistake in the main text.

Reviewer 2 Report

Comments and Suggestions for Authors

This review paper summarises the progress of therapeutic strategy which targets pathological proteins such as Tau and beta-amyloid ptrotein in Alzheier's disease. The authors describe growing evidence of proximity-induced pharmacology (PIP), and they also nicely introduced various kinds of small compounds in detail. Taken together, this review paper would be helpful for many readers working on the investigations of neurodegenerative diseases such as Alzheiemr's disease and Parkinson's disease.

Author Response

Thanks for your review.

Reviewer 3 Report

Comments and Suggestions for Authors

In this manuscript, the authors review current approaches exploring the use of proximity-induced pharmacology to treat amyloid-related diseases. The review is thorough and informative, though a number of grammatical errors, typographical errors, and other issues must be addressed, including items listed below:

Line 61: non-natives substrates- non-native?

Line 63: “particularities”, I think characteristics or properties would be more appropriate words

Line 71: Please explain why RTK activation would be useful in treating PD.  Even one sentence would be a great help to readers.

Line 78: Use of the term “warhead” is colloquial and distracting. Functional domains, interaction domains, interacting modules- find another term to describe these domains.

Line 83: In contrast

Line 97: a wild variety- change to wide variety

Line 142: “When changing the E3 ligase ligand from CRBN ligands to VHL ligands, - this phrase is a bit awkward.  When replacing the CRBN ligand to a VHL ligand..  or When changing the E3 ligase ligand from CRBN to that of VHL… When assessing the efficacy of degradation using different ligands, …

Line 209: “processes related to lysosomal pathway, mitochondria, neuroinflammation and α-synuclein accumulation.” The grammar is awkward here.  “processes related to lysosomal and mitochondrial function, neuroinflammation…” is a suggestion to fix this.

Line 224: “These aggregates are basically formed through the more than 35 glutamine residues of the polyglutamine (polyQ) tract” This is not clear.  “Htt aggregates form due to aggregation of an expanded polyglutamine domain in the pathogenic protein.” Or something similar would be easier to interpret.

Line 228: “besides” is too colloquial. “In addition” would be more appropriate

Line 232: “an inhibitor” not “and inhibitor” unless something else is intended here (it will not be clear to the reader how this molecule is designed)

Line 233: “its ubiquitination” is vague.  “target protein ubiquitination” is clearer

Line 234: “degradation of proteins in multiple targets”  this is unclear. “degradation of multiple protein targets”?

Line 235: “suffered” too colloquial– “failed due to “ would be more specific here

Line 239: “neurodegenerative diseases targets” disease targets, not diseases

Line 256: “the drug-like properties” pharmacokinetic issues?  What is intended here?

Line 280: “Neurodegenerative diseases, such as AD, PD, FTD, HD, and ALS have mutations that affect the autophagy mechanism.” What is intended here?  These diseases demonstrate defects in autophagy, but they may not be caused by genetic mutations. Also, how can targeting proteins for autophagy be successful if autophagy is impaired?

Line 295: proven, not proved

Line 302: Don’t use term “warhead” domain would be more appropriate, as above.

Line 317” showed a good” showed good penetration

Line 330: interact with mHtt, not “to”

Line 366: Distinguish, not recognize

Line 375: why is the specific sequence of the molecule described here if that kind of detail is left out of other descriptions? It is not clear what this detail adds to the review.

Line 397: Though may be more appropriate than however, otherwise the sentence is confusing.

Line 479: “In this sense, work is ongoing to improve to really impact the future perspectives of removing protein aggregates that cause neurodegeneration with PIP pharmacological approaches – This sentence is difficult to read and must be simplified. “Work is ongoing to continue to develop these technologies for clinical use” or something like that would be fine.

Line 482: amazing is too colloquial for a scientific review. “are promising technologies” or something like that would be more appropriate

Line 544: Why is the ribonucleotide targeting technology introduced in the Discussion? It should be placed just before it as another technology relevant to the manuscript.

The references were not numbered, so their relevance to specific statements in the manuscript could not be verified.

Comments on the Quality of English Language

As described above, grammatical and typographic errors need to be corrected, but the English is very good.

Author Response

Thank you for your thorough review. We have incorporated all your comments and suggestions. Here are our responses, point by point.

In this manuscript, the authors review current approaches exploring the use of proximity-induced pharmacology to treat amyloid-related diseases. The review is thorough and informative, though a number of grammatical errors, typographical errors, and other issues must be addressed, including items listed below:

  1. Line 61: non-natives substrates- non-native? Line 63: “particularities”, I think characteristics or properties would be more appropriate words. Line 78: Use of the term “warhead” is colloquial and distracting. Functional domains, interaction domains, interacting modules- find another term to describe these domains. Line 83: In contrast. Line 97: a wild variety- change to wide variety. Line 142: “When changing the E3 ligase ligand from CRBN ligands to VHL ligands, - this phrase is a bit awkward. When replacing the CRBN ligand to a VHL ligand..  or When changing the E3 ligase ligand Line 209: “processes related to lysosomal pathway, mitochondria, neuroinflammation and α-synuclein accumulation.” The grammar is awkward here.  “processes related to lysosomal and mitochondrial function, neuroinflammation…” is a suggestion to fix this. Line 224: “These aggregates are basically formed through the more than 35 glutamine residues of the polyglutamine (polyQ) tract” This is not clear.  “Htt aggregates form due to aggregation of an expanded polyglutamine domain in the pathogenic protein.” Or something similar would be easier to interpret. Line 228: “besides” is too colloquial. “In addition” would be more appropriate. Line 232: “an inhibitor” not “and inhibitor”. Line 233: “its ubiquitination” is vague.  “target protein ubiquitination” is clearer. Line 234: “degradation of proteins in multiple targets”  this is unclear. “degradation of multiple protein targets”? Line 235: “suffered” too colloquial– “failed due to “ would be more specific here. Line 239: “neurodegenerative diseases targets” disease targets, not diseases. Line 295: proven, not proved. Line 302: Don’t use term “warhead” domain would be more appropriate, as above. Line 317” showed a good” showed good penetration. Line 330: interact with mHtt, not “to”. Line 366: Distinguish, not recognize. Line 397: Though may be more appropriate than however, otherwise the sentence is confusing. Line 479: “In this sense, work is ongoing to improve to really impact the future perspectives of removing protein aggregates that cause neurodegeneration with PIP pharmacological approaches – This sentence is difficult to read and must be simplified. “Work is ongoing to continue to develop these technologies for clinical use” or something like that would be fine. Line 482: amazing is too colloquial for a scientific review. “are promising technologies” or something like that would be more appropriate.

Thank you very much for this thorough revision. All the typographical errors and grammar suggestions have been corrected in accordance with the reviewer's suggestions in the revised version of the draft.

  1. Line 71: Please explain why RTK activation would be useful in treating PD.  Even one sentence would be a great help to readers.

A sentence explaining the use of RTK activation as a treatment for PD has been included in the main text.

  1. Line 256: “the drug-like properties” pharmacokinetic issues?  What is intended here?

The paragraph has been modified to clarify this point. The modified text is: “Unfortunately, the authors did not extensively address the drug-like properties of SNIP-ERs molecules, such as membrane permeability and metabolic stability, reflecting the need of improvement of some pharmacokinetic properties to exploit the therapeutic potential of these molecules”.

  1. Line 280: “Neurodegenerative diseases, such as AD, PD, FTD, HD, and ALS have mutations that affect the autophagy mechanism.” What is intended here?  These diseases demonstrate defects in autophagy, but they may not be caused by genetic mutations. Also, how can targeting proteins for autophagy be successful if autophagy is impaired?

These diseases can have mutations that affect cellular autophagy mechanisms; consequently, the cleaning of these proteins is limited. AUTACs capture these proteins resistant to entering the autophagy system, directing them into it. The paragraph has been modified to clarify this point.

  1. Line 375: why is the specific sequence of the molecule described here if that kind of detail is left out of other descriptions? It is not clear what this detail adds to the review.

We have tried to describe all the molecules exemplified on this manuscript. In this case that the reviewer is pointing out, since we are talking about a peptide based HyT molecule we decided to describe the specific sequences needed to form the complete molecule, while remarking the main function of each part. The same has been done for other examples, such as for the CMA molecule targeting α-synuclein (line 342) or for the peptide-based Tau PROTAC (line 117).

  1. Line 544: Why is the ribonucleotide targeting technology introduced in the Discussion? It should be placed just before it as another technology relevant to the manuscript.

Along this manuscript we aimed to exemplify the different proximity-induced approaches described. Therefore, we intended to describe at least one example molecule of the approaches described targeting some proteins that are relevant in neurodegenerative diseases. Since there are no clear examples of RIBOTACs and PINADs already developed targeting neurodegenerative diseases, we decided just to introduce the concept and remark their therapeutic potential and limitations.

  1. The references were not numbered, so their relevance to specific statements in the manuscript could not be verified.

We appreciate the reviewer's comment. We have revised the references and numbering accordingly. However, the references were numbered and listed at the end of the manuscript.

Reviewer 4 Report

Comments and Suggestions for Authors

The review article titled "Proximity-induced pharmacology for amyloid related diseases", is well written. From protein degradative approaches to non-degradative approaches, various PIPs are well described in the context of various neurodegenerative diseases. However, there are a few minor corrections need to be done.

1. Page 2, Line no. 83: The sentence should start with "In", instead of just n.

2. Page 3, Line no. 97: To date, a wild variety of PROTACs have been designed (Replace wild with wide).

3. Page 4, Line no. 161-163: Arvinas’ tau-targeting PROTAC has demonstrated a reduction of insoluble aggregated tau protein in preclinical studies using a murine tauopathy model [24] (Replace this statement with Arvinas’ tau-targeting PROTAC has demonstrated a reduction of insoluble aggregated tau protein with respect to vehicle treatment, in preclinical studies using a murine tauopathy model [24]).

If the preclinical experiments are done on the same animal before and after treatment with PROTACs, and observed a reduction in the insoluble aggregates, then one can claim that PROTACs treatment reduces the insoluble aggregates in tauopathy model. If you observe the figure, they have compared Tau PROTAC treated mouse model with vehicle treated mouse model. Since proteasome cannot clear the tau aggregates itself, the effect observed in tau PROTAC treated mice is due the degradation of newly synthesized/ misfolded/hyperphosphorylated tau protein.

4. Page 5, line no. 199-200: Recently, a new set of GSK-3β has been synthetized (Replace with Recently, a new set of GSK-3β PROTACs has been synthetized).

Author Response

Thank you for your thorough review. We have incorporated all your comments and suggestions. Here are our responses, point by point.

The review article titled "Proximity-induced pharmacology for amyloid related diseases", is well written. From protein degradative approaches to non-degradative approaches, various PIPs are well described in the context of various neurodegenerative diseases. However, there are a few minor corrections need to be done.

  1. Page 2, Line no. 83: The sentence should start with "In", instead of just n. Page 3, Line no. 97: To date, a wild variety of PROTACs have been designed (Replace wild with wide). Page 5, line no. 199-200: Recently, a new set of GSK-3β has been synthetized (Replace with Recently, a new set of GSK-3β PROTACs has been synthetized).

Thank you very much. All the typographical errors and grammar suggestions have been corrected in accordance with the reviewer's suggestions in the revised version of the draft.

  1. Page 4, Line no. 161-163: Arvinas’ tau-targeting PROTAC has demonstrated a reduction of insoluble aggregated tau protein in preclinical studies using a murine tauopathy model [24] (Replace this statement with Arvinas’ tau-targeting PROTAC has demonstrated a reduction of insoluble aggregated tau protein with respect to vehicle treatment, in preclinical studies using a murine tauopathy model [24]).

The sentence has been modified in accordance with the reviewer's suggestions in the revised version of the draft. However, If the preclinical experiments are done on the same animal before and after treatment with PROTACs, and observed a reduction in the insoluble aggregates, then one can claim that PROTACs treatment reduces the insoluble aggregates in tauopathy model. If you observe the figure, they have compared Tau PROTAC treated mouse model with vehicle treated mouse model. Since proteasome cannot clear the tau aggregates itself, the effect observed in tau PROTAC treated mice is due the degradation of newly synthesized/ misfolded/hyperphosphorylated tau protein.